# An Empirical Study of simplicial Representation Learning with Wasserstein Distance

## Abstract

In this paper, we delve into the problem of simplicial representation learning utilizing the 1-Wasserstein distance on a tree structure (a.k.a., Tree-Wasserstein distance (TWD)), where TWD is defined as the L1 distance between two tree-embedded vectors. Specifically, we consider a framework for simplicial representation estimation employing a self-supervised learning approach based on SimCLR with a negative TWD as a similarity measure. In SimCLR, the cosine similarity with real-vector embeddings is often utilized; however, it has not been well studied utilizing L1-based measures with simplicial embeddings. A key challenge is that training the L1 distance is numerically challenging and often yields unsatisfactory outcomes, and there are numerous choices for probability models. Thus, this study empirically investigates a strategy for optimizing self-supervised learning with TWD and find a stable training procedure. More specifically, we evaluate the combination of two types of TWD (total variation and ClusterTree) and several simplicial models including the softmax function, the ArcFace probability model (Deng et al., 2019), and simplicial embedding (Lavoie et al., 2022). Moreover, we propose a simple yet effective Jeffrey divergence-based regularization method to stabilize the optimization. Through empirical experiments on STL10, CIFAR10, CIFAR100, and SVHN, we first found that the simple combination of softmax function and TWD can obtain significantly lower results than the standard SimCLR (non-simplicial model and cosine similarity). We found that the model performance depends on the combination of TWD and the simplicial model, and the Jeffrey divergence regularization usually helps model training. Finally, we inferred that the appropriate choice of combination of TWD and simplicial models outperformed cosine similarity based representation learning.

## 1 Introduction

Unsupervised learning is a widely studied topic, which includes the autoencoder (Kramer, 1991) and variational autoencoders (VAE) (Kingma & Welling, 2013). Self-supervised learning algorithms including SimCLR (Chen et al., 2020b), Bootstrap Your Own Latent (BYOL) (Grill et al., 2020) and MoCo (Chen et al., 2020b; He et al., 2020) can also be regarded as unsupervised learning methods. The key idea of a self-supervised algorithm is to adopt contrastive learning, in which two data points are systematically generated from a common data source, and lower representations are found by maximizing the similarity of these two generated input vectors while minimizing the similarity between a target and negative samples. For example, we generate two images by independently applying image transformation such as rotation and cropping.

In terms of contrastive learning loss functions, the InfoNCE (Oord et al., 2018) is widely utilized. It is a powerful pretraining method because self-supervised learning does not require label information and can be trained with several data points. Most previous studies have focused on obtaining real-valued vector representations. However, a few studies have learned lower representations on a simplex (i.e., simplicial representation), including simplicial embedding (Lavoie et al., 2022).

Wasserstein distance, a widely adopted optimal transport-based distance for measuring distributional difference, is useful in various machine learning tasks including generative adversarial networks (Arjovsky et al., 2017), document classification (Kusner et al., 2015; Sato et al., 2022), and image matching (Liu et al., 2020; Sarlin et al., 2020), to name a few. The Wasserstein distance is also

known as the earth mover's distance (EMD) in the computer vision community and the word mover's distance (WMD) in the natural language processing community.

In this paper, we consider the problem of learning a simplicial representation by utilizing a self-supervised framework with 1-Wasserstein distance with tree metric (i.e., Tree-Wasserstein Distance (TWD)). TWD includes the sliced Wasserstein distance and total variation as special cases, and it can be represented by the L1 distance between tree-embedded vectors. Because TWD is given as a non-differentiable function, learning simplicial representation utilizing backpropagation of TWD is challenging. Moreover, there exist several probability representations exist, and it is difficult to determine which one is more suitable for simplicial representation learning. Hence, we investigate a combination of probability models and the structure of TWD. More specifically, we consider the total variation and the ClusterTree for TWD, where we show that the total variation is equivalent to the robust variant of TWD. Then, we propose the utilization of softmax, ArcFace based probability model (Deng et al., 2019), and the simplicial embedding (SEM) (Lavoie et al., 2022). Moreover, to stabilize the training, we propose a Jeffrey divergence based regularization. Through self-supervised learning experiments, we found that the standard softmax formulation with backpropagation yield poor results. For total variation, the combination of the ArcFace-based model performs well. In contrast, SEM is suitable for ClusterTree, whereas the ArcFace-based models achieve modest performance. Moreover, the proposed regularization significantly outperforms when compared to its non-regularized counterparts.

**Contribution:** The contributions are summarized below:

- We investigated the combination of probability models and TWD (total variation and ClusterTree). Then, we found that the ArcFace model with prior information is suited for total variation, while SEM (Lavoie et al., 2022) is suited for ClusterTree models.

- We proposed a robust variant of TWD (RTWD) and showed that RTWD is equivalent to total variation.

- We proposed the Jeffrey divergence regularization for TWD minimization; the regularization significantly stabilized the training.

- We demonstrated that the combination of TWD and simplicial models can obtain better performance in self-supervised training for CIFAR10, STL10, and SVHN compared to the cosine similarity and real-valued representation combination, while the performance of CIFAR100 can be improved further in future.

## 2 RELATED WORK

The proposed method involved unsupervised representation learning and optimal transportation.

**Unsupervised Representation Learning:** Representation learning is an important research topic in machine learning and involves several methods. The autoencoder (Kramer, 1991) and its variational version (Kingma & Welling, 2013) are widely employed in unsupervised representation learning methods. Current mainstream self-supervised learning (SSL) approaches are based on a cross-view prediction framework (Becker & Hinton, 1992), and contrastive learning has emerged as a prominent self-supervised learning paradigm.

In SSL, a model learns by contrasting positive samples (similar instances) with negative samples (dissimilar instances), utilizing methods such as SimCLR (Chen et al., 2020a), BYOL (Grill et al., 2020), and MoCo (Chen et al., 2020b; He et al., 2020). These methods employ data augmentation and similarity metrics to encourage the model to project similar instances close together while pushing dissimilar ones apart. This approach has demonstrated effectiveness across various domains, including computer vision and natural language processing, thereby enabling learning without explicit labels. SimCLR employs InfoNCE loss (Oord et al., 2018). After SimCLR has proposed, several alternative approaches have been proposed, such as Barlow's twin (Zbontar et al., 2021) and BYOL (Grill et al., 2020). Barlow Twins loss function attempts to maximize the correlation between positive pairs while minimizing the cross-correlation with negative samples. Barlow Twins is closely related to the Hilbert–Shmidt independence criterion, a kernel-based independence measure (Gretton et al., 2005; Tsai et al., 2021).

These approaches mainly focus on learning lower-dimensional representations with real-valued vector embeddings, and cosine similarity is adopted as a similarity measure in contrastive learning. Recently, Lavoie et al. (2022) proposed the simplicial embedding (SEM), which comprises multiple concatenated softmax functions and learns high-dimensional sparse nonnegative representations. This significantly improves the classification accuracy.

**Divergence and optimal transport:** Measuring the divergence between two probability distributions is a fundamental research problem in machine learning, and it has been utilized for various computer vision and natural language processing applications, including document classification (Kusner et al., 2015; Sato et al., 2022) and image matching (Sarlin et al., 2020). One widely adopted divergence measure is Kullback–Leibler (KL) divergence (Cover & Thomas, 2012). However, KL divergence can diverge to infinity when the supports of the two input probability distributions do not overlap.

The Wasserstein distance, or, as is known in the computer vision community, the EMD, can handle the difference in supports in probability distributions. The Wasserstein distance was computed by linear programming. Moreover, in addition to handling different supports, one of the key advantages of Wasserstein distance over KL is that it can identify matches between data samples. For example, Sarlin et al. (2020) proposed SuperGlue, leveraging optimal transport for correspondence determination in local feature sets. Semantic correspondence investigated optimal transport by Liu et al. (2020). In NLP, Kusner et al. (2015) introduced the WMD, a Wasserstein distance pioneer in textual similarity tasks, widely utilized, including text generation evaluation (Zhao et al., 2019). Sato et al. (2022) further studied the property of WMD thoroughly. Another interesting approach is the word rotator distance (WRD) (Yokoi et al., 2020), which utilizes the normalized norm of word vectors as a simplicial representation and significantly improves WMD's performance. However, these methods incur cubic-order computational costs, rendering them unsuitable for extensive distribution comparison tasks.

To speed up EMD and Wasserstein distance computation, Cuturi (2013) introduced the Sinkhorn algorithm, solving the entropic regularized optimization problem and achieving quadratic order Wasserstein distance computation ($O(\bar{n}^2)$), where $\bar{n}$ is the number of data points. Moreover, because the optimal solution of the Sinkhorn algorithm can be obtained by an iterative algorithm, it can easily be incorporated into deep-learning applications, which makes optimal transport widely applicable to deep-learning applications. One limitation of the Sinkhorn algorithm is that it still requires quadratic-time computation, and the final solution depends highly on the regularization parameter.

An alternative approach is the sliced Wasserstein distance (SWD) (Rabin et al., 2011; Kolouri et al., 2016), which solves the optimal transport problem within a projected one-dimensional subspace. Given 1D optimal transport's sortable nature, SWD offers $O(\bar{n} \log \bar{n})$ computations. SWD's extensions include the generalized sliced Wasserstein distance for multidimensional cases (Kolouri et al., 2019), the max-sliced Wasserstein distance, which determines the optimal transport-enhancing 1D subspace (Mueller & Jaakkola, 2015; Deshpande et al., 2019), and the subspace-robust Wasserstein distance (Paty & Cuturi, 2019).

The 1-Wasserstein distance with a tree metric (also known as the Tree-Wasserstein Distance (TWD)) is a generalization of the sliced Wasserstein distance and total variation (Indyk & Thaper, 2003; Evans & Matsen, 2012; Le et al., 2019). Moreover, TWD is also known as the UniFrac distance (Lozupone & Knight, 2005), which is assumed to have a phylogenetic tree beforehand. An important property of TWD is that TWD has an analytical solution with the L1 distance of tree-embedded vectors. Originally, TWD was studied in theoretical computer science, known as the QuadTree algorithm (Indyk & Thaper, 2003), and was recently extended to unbalanced TWD (Sato et al., 2020; Le & Nguyen, 2021), supervised Wasserstein training (Takezawa et al., 2021), and tree barycenters (Takezawa et al., 2022) in the ML community. These approaches focused on approximating the 1-Wasserstein distance through tree construction, often utilizing constant-edge weights. Backurs et al. (2020) introduced FlowTree, amalgamating QuadTree and cost matrix methods, outperforming QuadTree. They guaranteed that QuadTree and FlowTree approximated the nearest neighbors employing the 1-Wasserstein distance. Dey & Zhang (2022) proposed L1-embedding for approximating 1-Wasserstein distance for persistence diagrams. Yamada et al. (2022) proposed a computationally efficient tree weight estimation technique for TWD and empirically demonstrated that

TWD can have comparable performance to the Wasserstein distance while the computational speed is several orders of magnitude faster than the original Wasserstein distance.

Most existing studies on TWD focus on tree construction (Indyk & Thaper, 2003; Le et al., 2019; Takezawa et al., 2021) and edge weight estimation (Yamada et al., 2022). Takezawa et al. (2022) proposed a Barycenter method based on TWD where the set of simplicial representationes are given. Frogner et al. (2015) considered utilizing the Wasserstein distance for multilabel classification. Moreover, Toyokuni et al. (2021) utilized TWD for multilabel classification. These studies focused on supervised learning employing softmax as the probability model. Here, we investigated the Wasserstein distance employing a self-supervised learning framework and evaluated various probability models.

## 3 BACKGROUND

### 3.1 SELF-SUPERVISED LEARNING WITH SIMCLR

Let us denote $\boldsymbol{x} \in \mathbb{R}^d$ as an input vector, and let us have $n$ input vectors $\{\boldsymbol{x}_i\}_{i=1}^n$. Now we define the data transformation functions $\boldsymbol{u}^{(1)} = \boldsymbol{\phi}_1(\boldsymbol{x}) \in \mathbb{R}^d$ and $\boldsymbol{u}^{(2)} = \boldsymbol{\phi}_2(\boldsymbol{x}) \in \mathbb{R}^d$, respectively. The neural network model is denoted by $\boldsymbol{f}_{\boldsymbol{\theta}}(\boldsymbol{u}) \in \mathbb{R}^{d_{\text{out}}}$, where $\boldsymbol{\theta}$ is a learning parameter. Then, SimCLR attempts to train the model by closing $\boldsymbol{u}^{(1)}$ and $\boldsymbol{u}^{(2)}$ to be close, where $\boldsymbol{u}^{(1)}$ and/or $\boldsymbol{u}^{(2)}$ are dissimilar to $\boldsymbol{u}'$, which is a negative sample.

To this end, the infoNCE (Oord et al., 2018) is employed in the SimCLR model:

$$\ell_{\text{infoNCE}}(\boldsymbol{z}_i, \boldsymbol{z}_j) = -\log \frac{\exp(\text{sim}(\boldsymbol{z}_i, \boldsymbol{z}_j)/\tau)}{\sum_{k=1}^{2R} \delta_{k \neq i} \exp(\text{sim}(\boldsymbol{z}_i, \boldsymbol{z}_k)/\tau)},$$

where $R$ is the batch size and $\text{sim}(\boldsymbol{u}, \boldsymbol{u}')$ is a similarity function that takes a higher positive value if $\boldsymbol{u}$ and $\boldsymbol{u}'$ are similar, whereas it takes a smaller positive value or negative value when $\boldsymbol{u}$ and $\boldsymbol{u}'$ are dissimilar, $\tau$ is the temperature parameter, and $\delta_{k \neq i}$ is a delta function that takes a value of 1 when $k \neq i$ and value 0 otherwise. In SimCLR, the parameter is learned by minimizing the infoNCE loss. Hence, the first term of the infoNCE loss makes $\boldsymbol{z}_i$ and $\boldsymbol{z}_j$ similar. The second term is a log sum exp function and is a softmax function. Because we attempt to minimize the maximum similarity between the input $\boldsymbol{z}_i$ and its negative samples, we can make $\boldsymbol{z}_i$ and its negative samples dissimilar utilizing the second term.

$$\widehat{\boldsymbol{\theta}} := \underset{\boldsymbol{\theta}}{\text{argmin}} \sum_{i=1}^n \ell_{\text{infoNCE}}(\boldsymbol{z}_i^{(1)}, \boldsymbol{z}_i^{(2)}).$$

where $\boldsymbol{z}_i^{(1)} = \boldsymbol{f}_{\boldsymbol{\theta}}(\boldsymbol{u}_i^{(1)})$ and $\boldsymbol{z}_i^{(2)} = \boldsymbol{f}_{\boldsymbol{\theta}}(\boldsymbol{u}_i^{(2)})$, respectively.

### 3.2 $p$-WASSERSTEIN DISTANCE

The $p$-Wasserstein distance between two discrete measures $\mu = \sum_{i=1}^{\bar{n}} a_i \delta_{\boldsymbol{x}_i}$ and $\mu' = \sum_{j=1}^{\bar{m}} a_j' \delta_{\boldsymbol{y}_j}$ is given by

$$\mathcal{W}_p(\mu, \mu') = \left( \min_{\boldsymbol{\Pi} \in U(\mu, \mu')} \sum_{i=1}^{\bar{n}} \sum_{j=1}^{\bar{m}} \pi_{ij} d(\boldsymbol{x}_i, \boldsymbol{y}_j)^p \right)^{1/p},$$

where $U(\mu, \mu')$ denotes the set of the transport plans and $U(\mu, \mu') = \{\boldsymbol{\Pi} \in \mathbb{R}_+^{\bar{n} \times \bar{m}} : \boldsymbol{\Pi} \mathbf{1}_{\bar{m}} = \boldsymbol{a}, \boldsymbol{\Pi}^\top \mathbf{1}_{\bar{n}} = \boldsymbol{a}'\}$. The Wasserstein distance is computed utilizing a linear program. However, because it includes an optimization problem, the computation of Wasserstein distance for each iteration is computationally expensive.

One of the alternative approaches is to use the entropic regularization (Cuturi, 2013). If we consider the 1-Wasserstein distance, the entropic regularized variant is given as

$$\min_{\boldsymbol{\Pi} \in U(\boldsymbol{a}, \boldsymbol{a}')} \sum_{i=1}^{\bar{n}} \sum_{j=1}^{\bar{m}} \pi_{ij} d(\boldsymbol{x}_i, \boldsymbol{y}_j) + \lambda \sum_{i=1}^{\bar{n}} \sum_{j=1}^{\bar{m}} \pi_{ij}(\log(\pi_{ij}) - 1).$$

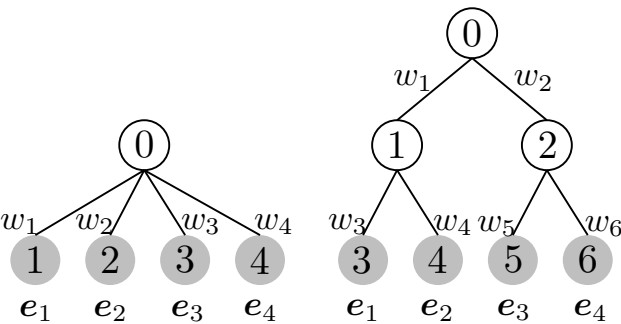

Figure 1: Left tree corresponds to the total variation if we set the weight as $w_i = \frac{1}{2}, \forall i$. Right tree is a ClusterTree (2 class).

This optimization problem can be solved efficiently by using the Sinkhorn algorithm at a computational cost of $O(\bar{n}\bar{m})$. More importantly, the solution of the Sinkhorn algorithm is given as a series of matrix multiplications; the Sinkhorn algorithm is widely employed for deep learning algorithms.

### 3.3 1-Wasserstein distance with tree metric (Tree-Wasserstein Distance)

Another 1-Wasserstein distance is based on trees (Indyk & Thaper, 2003; Le et al., 2019). The 1-Wasserstein distance with tree metric is defined as the L1 distance between the two probability distributions $\mu = \sum_j a_j \delta_{\boldsymbol{x}_j}$ and $\mu' = \sum_j a'_j \delta_{\boldsymbol{x}_j}$:

$$W_{\mathcal{T}}(\mu, \mu') = \|\mathrm{diag}(\boldsymbol{w})\boldsymbol{B}(\boldsymbol{a} - \boldsymbol{a}')\|_1 = \|\mathrm{diag}(\boldsymbol{w})\boldsymbol{B}\boldsymbol{a} - \mathrm{diag}(\boldsymbol{w})\boldsymbol{B}\boldsymbol{a}'\|_1,$$

where $\boldsymbol{B} \in \{0, 1\}^{N_{\text{node}} \times N_{\text{leaf}}}$ and $\boldsymbol{w} \in \mathbb{R}_+^{N_{\text{node}}}$. Here, we consider a tree with a depth of one and a ClusterTree, as illustrated in Figure 1. If all the edge weights $w_1 = w_2 = \ldots = w_N = \frac{1}{2}$ in the left panel of Figure 1, the $\boldsymbol{B}$ matrix is given as $\boldsymbol{B} = \boldsymbol{I}$. Substituting this result into the TWD, we have

$$W_{\mathcal{T}}(\mu, \mu') = \frac{1}{2}\|\boldsymbol{a} - \boldsymbol{a}'\|_1 = \|\boldsymbol{a} - \boldsymbol{a}'\|_{\text{TV}}.$$

Thus, the total variation is a special case of TWD. In this setting, the shortest path distance in the tree corresponds to the Hamming distance. Note that Raginsky et al. (2013) also assert that the 1-Wasserstein distance with the Hamming metric $d(x, y) = \delta_{x \neq y}$ is equivalent to the total variation (Proposition 3.4.1).

The key advantage of the tree-based approach is that the Wasserstein distance is provided in closed form, which is computationally very efficient. Note that a chain is included as a special case of a tree. Thus, the widely employed sliced Wasserstein distance is included as a special case of TWD. Moreover, it has been empirically reported that TWD and Sinkhorn-based approaches perform similarly in multi-label classification tasks (Toyokuni et al., 2021).

## 4 Self-supervised Learning with 1-Wasserstein Distance

In this section, we first formulate self-supervised learning utilizing the TWD. We then introduce ArcFace-based probability models and Jeffrey divergence regularization.

### 4.1 SimCLR with Tree Wasserstein Distance

$\boldsymbol{a}$ and $\boldsymbol{a}'$ are the simplicial embedding vectors of $\boldsymbol{x}$ and $\boldsymbol{x}'$ (i.e., $\boldsymbol{1}^\top \boldsymbol{a} = 1$ and $\boldsymbol{1}^\top \boldsymbol{a}' = 1$) with $\mu = \sum_j a_j \delta_{\boldsymbol{e}_j}$ and $\mu' = \sum_j a'_j \delta_{\boldsymbol{e}_j}$, respectively. Here, $\boldsymbol{e}_j$ is the virtual embedding corresponding to $a_j$ or $a'_j$, where $\boldsymbol{e}$ is not available in out problem setup in general. Then, we adopt the negative Wasserstein distance between $\mu$ and $\mu'$ as the similarity score for SimCLR:

$$\mathrm{sim}(\mu, \mu') = -W_{\mathcal{T}}(\mu, \mu') = -\|\mathrm{diag}(\boldsymbol{w})\boldsymbol{B}\boldsymbol{a} - \mathrm{diag}(\boldsymbol{w})\boldsymbol{B}\boldsymbol{a}'\|_1.$$

We cannot determine the tree structure or its weight; we assume that $\boldsymbol{B}$ and $\boldsymbol{w}$ are given. We considered the total variation and ClusterTree cases (see Figure 1).

This study attempts to estimate a simplicial representation estimation by adopting self-supervised learning:

$$\widehat{\boldsymbol{\theta}} := \operatorname*{argmin}_{\boldsymbol{\theta}} \sum_{i=1}^{n} \left( W_{\mathcal{T}}(\mu_i^{(1)}, \mu_j^{(2)})\tau^{-1} + \log \sum_{k=1}^{2N} \delta_{k \neq i} \exp(-W_{\mathcal{T}}(\mu_i^{(1)}, \mu_j^{(2)})\tau^{-1}) \right),$$

where $\tau > 0$ is the temperature parameter for the InfoNCE loss. We can certainly utilize Barlow Twins instead of infoNCE for simplicial representation estimation.

## 4.2 ROBUST VARIANT OF TREE WASSERSTEIN DISTANCE

In our setup, it is hard to estimate the tree structure $\boldsymbol{B}$ and the edge weight $\boldsymbol{w}$, because the embedding vectors $\boldsymbol{e}_1, \boldsymbol{e}_2, \ldots, \boldsymbol{e}_{d_{\text{out}}}$ is not available. To deal with the problem, we consider the robust estimation of Wasserstein distance such as the subspace robust Wasserstein distance (SRWD) (Paty & Cuturi, 2019) for TWD. Specifically, with given $\boldsymbol{B}$, we propose the robust TWD (RTWD) as follows:

$$\text{RTWD}(\mu, \mu') = \frac{1}{2} \min_{\boldsymbol{\Pi} \in U(\boldsymbol{a}, \boldsymbol{a}')} \max_{\boldsymbol{w} \text{ s.t. } \boldsymbol{B}^\top \boldsymbol{w} = \boldsymbol{1} \text{ and } \boldsymbol{w} \geq \boldsymbol{0}} \sum_{i=1}^{N_{\text{leafs}}} \sum_{j=1}^{N_{\text{leafs}}} \pi_{ij} d_{\mathcal{T}}(\boldsymbol{e}_i, \boldsymbol{e}_j),$$

where $d_{\mathcal{T}}(\boldsymbol{e}_i, \boldsymbol{e}_j)$ is the shortest path distance between $\boldsymbol{e}_i$ and $\boldsymbol{e}_j$, where $\boldsymbol{e}_i$ and $\boldsymbol{e}_j$ are embedded on a tree $\mathcal{T}$.

**Proposition 1** *The robust variant of TWD (RTWD) is equivalent to total variation:*

$$\text{RTWD}(\mu, \mu') = \|\boldsymbol{a} - \boldsymbol{a}'\|_{\text{TV}},$$

*where $\|\boldsymbol{a} - \boldsymbol{a}'\|_{\text{TV}} = \frac{1}{2}\|\boldsymbol{a} - \boldsymbol{a}'\|_1$ is the total variation.*

Based on this proposition, the RTWD is equivalent to a total variation and it does not depend on the tree structure $\boldsymbol{B}$. That is, if we do not have a prior information about tree structure, using the total variation is a reasonable choice.

## 4.3 PROBABILITY MODELS

Several probability models were adopted here.

**Softmax:** The softmax function for simplicial representation is given by

$$\boldsymbol{a}_{\boldsymbol{\theta}}(\boldsymbol{x}) = \text{Softmax}(\boldsymbol{f}_{\boldsymbol{\theta}}^{(\ell)}(\boldsymbol{x})),$$

where $\boldsymbol{f}_{\boldsymbol{\theta}}(\boldsymbol{x}_i)$ is a neural network model.

**Simplicial Embedding:** Lavoie et al. (2022) proposed a simple yet efficient simplicial embedding method. We assume that the output dimensionality of the output of the neural network model is $d_{\text{out}}$. Then, SEM applies the softmax function to each $V$-dimsensional vector of $\boldsymbol{f}_{\boldsymbol{\theta}}(\boldsymbol{x})$, where we have $L = d_{\text{out}}/V$ probability vectors. The $\ell$-th softmax function is defined as follows.

$$\boldsymbol{a}_{\boldsymbol{\theta}}(\boldsymbol{x}) = [\boldsymbol{a}_{\boldsymbol{\theta}}^{(1)}(\boldsymbol{x})^\top, \boldsymbol{a}_{\boldsymbol{\theta}}^{(2)}(\boldsymbol{x})^\top, \ldots, \boldsymbol{a}_{\boldsymbol{\theta}}^{(L)}(\boldsymbol{x})^\top]^\top \text{ with } \boldsymbol{a}_{\boldsymbol{\theta}}^{(\ell)}(\boldsymbol{x}) = \text{Softmax}(\boldsymbol{f}_{\boldsymbol{\theta}}^{(\ell)}(\boldsymbol{x}))/L,$$

where $\boldsymbol{f}_{\boldsymbol{\theta}}^{(\ell)}(\boldsymbol{x})) \in \mathbb{R}^V$ is the $\ell$-th block of a neural network model. We normalize the softmax function by $L$ because $\boldsymbol{a}_{\boldsymbol{\theta}}(\boldsymbol{x})$ must satisfy the sum to one constraint for Wasserstein distance.

**ArcFace model (AF):** We proposed employing the ArcFace probability model (Deng et al., 2019).

$$\boldsymbol{a}_{\boldsymbol{\theta}}(\boldsymbol{x}) = \text{Softmax}(\boldsymbol{K}^\top \boldsymbol{f}_{\boldsymbol{\theta}}(\boldsymbol{x})/\eta) = \left( \frac{\exp(\boldsymbol{k}_1^\top \boldsymbol{f}_{\boldsymbol{\theta}}(\boldsymbol{x})/\eta)}{\sum_{k=1}^{d_{\text{out}}} \exp(\boldsymbol{k}_k^\top \boldsymbol{f}_{\boldsymbol{\theta}}(\boldsymbol{x})/\eta)}, \ldots, \frac{\exp(\boldsymbol{k}_{d_{\text{out}}}^\top \boldsymbol{f}_{\boldsymbol{\theta}}(\boldsymbol{x})/\eta)}{\sum_{k=1}^{d_{\text{out}}} \exp(\boldsymbol{k}_k^\top \boldsymbol{f}_{\boldsymbol{\theta}}(\boldsymbol{x})/\eta)} \right),$$

where $\boldsymbol{K} = [\boldsymbol{k}_1, \boldsymbol{k}_2, \ldots, \boldsymbol{k}_{d_{\text{out}}}] \in \mathbb{R}^{d_{\text{out}} \times d_{\text{prob}}}$ is a learning parameter, $\boldsymbol{f}_{\boldsymbol{\theta}}(\boldsymbol{x})$ is a normalized output of a model ($\boldsymbol{f}_{\boldsymbol{\theta}}(\boldsymbol{x})^\top \boldsymbol{f}_{\boldsymbol{\theta}}(\boldsymbol{x}) = 1$), and $\eta$ is a temperature parameter. Note that AF has a similar structure of attention in transformers (Bahdanau et al., 2014; Vaswani et al., 2017). The key difference from the original attention is normalizing the key matrix $\boldsymbol{K}$ and the query vector $\boldsymbol{f}_{\boldsymbol{\theta}}(\boldsymbol{x})$.

**AF with Positional Encoding:** The AF can add one more linear layer and then apply the softmax function; the performance can generally be similar to the standard softmax function. Here, we propose replacing the key matrix with a normalized positional encoding matrix $\boldsymbol{k}_i^\top \boldsymbol{k}_i = 1, \forall i$):

$$\boldsymbol{k}_i = \frac{\bar{\boldsymbol{k}}_i}{\|\bar{\boldsymbol{k}}_i\|_2}, \ \ \bar{\boldsymbol{k}}_i^{(2j)} = \sin(i/10000^{2j/d_{\text{out}}}), \ \ \bar{\boldsymbol{k}}_i^{(2j+1)} = \cos(i/10000^{2j/d_{\text{out}}}).$$

**AF with Discrete Cosine Transform Matrix:** Another natural approach would be to utilize an orthogonal matrix as $\boldsymbol{K}$. Therefore, we proposed adopting a discrete cosine transform (DCT) matrix as $\boldsymbol{K}$. The DCT matrix is expressed as

$$\boldsymbol{k}_i^{(j)} = \begin{cases} 1/\sqrt{d_{\text{out}}} & (i=0) \\ \sqrt{\frac{2}{d_{\text{out}}}} \cos \frac{\pi(2j+1)i}{2d_{\text{out}}} & (1 \le i \le d_{\text{out}}) \end{cases}.$$

One of the contributions of this study is that combining positional encoding and DCT matrix with the ArcFace model significantly boosts performance, whereas the standard ArcFace model performs similarly to the simple softmax function.

## 4.4 JEFFREY-DIVERGENCE REGULARIZATION

We empirically observed that optimizing the loss function above is extremely challenging. One problem is that L1 distance is not differentiable at 0. Another issue is that the softmax function has many degrees of freedom and cannot utilize the cost function in our framework. Figure 2 illustrates the learning curve for the standard optimization adopting the softmax function model.

To stabilize the optimization, we proposed the Jeffrey divergence regularization, which is the upper bound of the square of the 1-Wasserstein distance.

**Proposition 2** *For $\boldsymbol{B}^\top \boldsymbol{w} = \boldsymbol{1}$ and the probability vectors $\boldsymbol{a}_i$ and $\boldsymbol{a}_j$, we have*

$$W_{\mathcal{T}}^2(\mu_i, \mu_j) \le \text{JD}(\text{diag}(\boldsymbol{w})\boldsymbol{B}\boldsymbol{a}_i \| \text{diag}(\boldsymbol{w})\boldsymbol{B}\boldsymbol{a}_j),$$

*where $\text{JD}(\text{diag}(\boldsymbol{w})\boldsymbol{B}\boldsymbol{a}_i \| \text{diag}(\boldsymbol{w})\boldsymbol{B}\boldsymbol{a}_j)$ is a Jeffrey divergence.*

This result indicates that minimizing the symmetric KL divergence (i.e., Jeffrey divergence) can minimize the tree-Wasserstein distance. Because the Jeffrey divergence is smooth, computation of the gradient of the upper bound is easier.

We propose the following regularized version of InfoNCE.

$$\bar{\ell}_{\text{InfoNCE}}(\boldsymbol{a}_i^{(1)}, \boldsymbol{a}_j^{(2)}) = \tau^{-1} W_{\mathcal{T}}(\boldsymbol{a}_i^{(1)}, \boldsymbol{a}_j^{(2)}) + \log \sum_{k=1}^{2N} \delta_{k \ne i} \exp(-\tau^{-1} W_{\mathcal{T}}(\boldsymbol{a}_i^{(1)}, \boldsymbol{a}_j^{(2)}))$$

$$+ \lambda \text{JD}(\text{diag}(\boldsymbol{w})\boldsymbol{B}\boldsymbol{a}_i^{(1)} \| \text{diag}(\boldsymbol{w})\boldsymbol{B}\boldsymbol{a}_j^{(2)}).$$

Note that Frogner et al. (2015) considered a multilabel classification problem utilizing a regularized Wasserstein loss. They proposed utilizing the Kullback–Leibler divergence-based regularization to stabilize the training. We derive the Jeffrey divergence from the TWD, which includes a simple KL divergence-based regularization as a special case. Moreover, we proposed employing JD regularization for self-supervised learning frameworks, which has not been extensively studied.

## 5 EXPERIMENTS

This section evaluates SimCLR methods with different probability models utilizing STL10, CIFAR10, CIFAR100, and SVHN datasets.

## 5.1 SETUP

For all experiments, we employed the Resnet18 model with an output dimension of ($d_{\text{out}} = 256$) and coded all methods based on a standard SimCLR implementation [1]. We used the Adam optimizer and

---

[1]`https://github.com/sthalles/SimCLR`

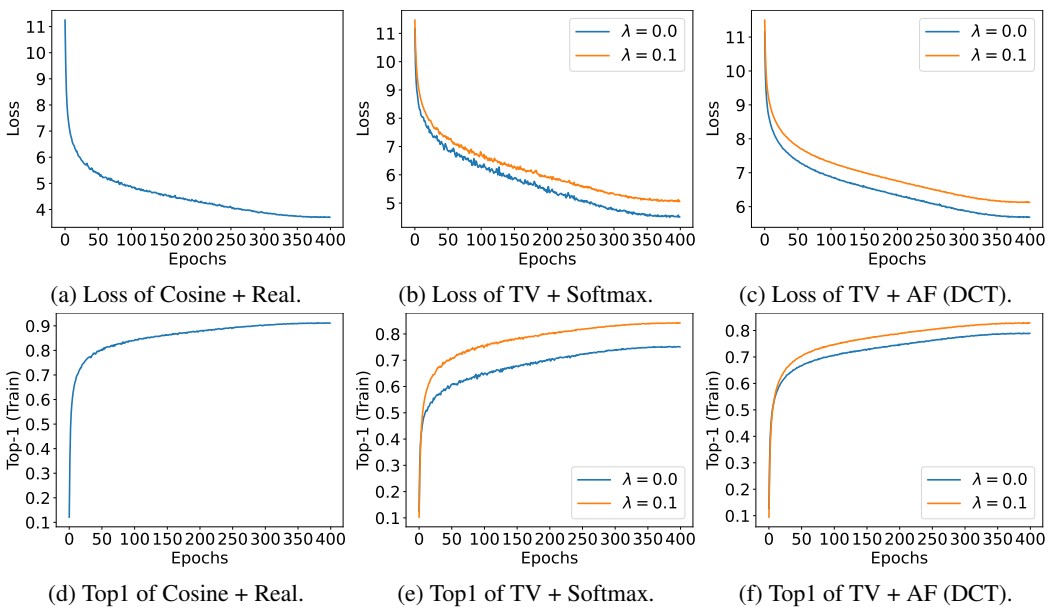

(a) Loss of Cosine + Real.  (b) Loss of TV + Softmax.  (c) Loss of TV + AF (DCT).

(d) Top1 of Cosine + Real.  (e) Top1 of TV + Softmax.  (f) Top1 of TV + AF (DCT).

Figure 2: The infoNCE loss and Top-1 (Train) comparisons on STL10 dataset.

set the learning rate as 0.0003, weight decay parameter as 1e-4, adn temperature $\tau$ as 0.07, respectively. For the proposed method, we compared two variants of TWD: total variation and ClusterTree (See Figure 1). As a model, we evaluated the conventional softmax function, attention model (AF), and simplicial embedding (SEM) (Lavoie et al., 2022) and set the temperature parameter $\tau = 0.1$ for all experiments. For SEM, we set $L = 16$ and $V = 16$.

We also evaluated JD regularization, where we set the regularization parameter $\lambda = 0.1$ for all experiments. For reference, we compared the cosine similarity as a similarity function of SimCLR. For all approaches, we utilized the KNN classifier of the scikit-learn package[2], where the number of nearest neighbor was set to $K = 50$ and utilized the L1 distance for Wasserstein distances and cosine similarity for non-simplicial models. For all experiments, we utilized A6000 GPUs. We ran all experiments three times and reported the average scores.

## 5.2 PERFORMANCE COMPARISON

Figure 2 illustrates the training loss and top-1 accuracy for three methods: cosine + real-valued embedding, TV + Softmax, and TV + AF (DCT). This experiment revealed that the convergence speed of the loss is nearly identical across all methods. Regarding the training top-1 accuracy, Cosine + Real-valued embedding achieves the highest accuracy, followed by the Softmax function, and AF (DCT) lags. This behavior is expected because real-valued embeddings offer the most flexibility, Softmax comes next, and AF models have the least freedom. For all methods based on Tree Wasserstein Distance (TWD), it is evident that JD regularization significantly aids the training process, especially in the case of the Softmax function. However, for AF (DCT), the improvement was relatively small. This is likely because AF (DCT) can also be considered a form of regularization.

Table 1 presents the experimental results for test classification accuracy using KNN. The first observation is that the simple implementation of the conventional Softmax function performs very poorly (the performance is approximately 10 points lower) compared with the cosine similarity. As expected, AF has just one more layer than the simple Softmax model, performing similarly to Softmax. Compared with Softmax and AF, AF (PE) and AF (DCT) significantly improved the classification accuracy for the total variation and ClusterTree cases. However, for the ClusterTree case, AF (PE) achieved a better classification performance, whereas the AF (DCT) improvement over the softmax model was limited. In the ClusterTree case, SEM significantly improves with a combination of ClusterTree and regularization. Overall, the proposed methods performed better than cosine

[2]https://scikit-learn.org/stable/modules/generated/sklearn.neighbors.KNeighborsClassifier.html

Table 1: KNN classification result with Resnet18 backbone. In this experiment, we set the number of neighbors as $K = 50$ and computed the averaged classification accuracy over three runs. Note that the Wasserstein distance with $(\boldsymbol{B} = \boldsymbol{I}_{d_{\text{out}}})$ is equivalent to a total variation.

| Similarity Function | simplicial Model | STL10 | CIFAR10 | CIFAR100 | SVHN |
|---|---|---|---|---|---|
| Cosine Similarity | N/A | **75.77** | **67.39** | **32.06** | 76.35 |
| | Softmax | 70.12 | 63.20 | 26.88 | 74.46 |
| | SEM | 71.33 | 61.13 | 26.08 | 74.28 |
| | AF (DCT) | 72.95 | 65.92 | 25.96 | **76.51** |
| TWD (TV) | Softmax | 65.54 | 59.72 | 26.07 | 72.67 |
| | SEM | 65.35 | 56.56 | 24.31 | 73.36 |
| | AF | 65.61 | 60.92 | 26.33 | 75.01 |
| | AF (PE) | 71.71 | 64.68 | 26.38 | 76.44 |
| | AF (DCT) | 73.28 | 67.03 | 25.85 | 77.62 |
| | Softmax + JD | 72.64 | 67.08 | **27.82** | 77.69 |
| | SEM + JD | 71.79 | 63.60 | 26.14 | 75.64 |
| | AF + JD | 72.64 | 67.15 | 27.45 | 78.00 |
| | AF (PE) + JD | 74.47 | 67.28 | 27.01 | 78.12 |
| | AF (DCT) + JD | **76.28** | **68.60** | 26.49 | **79.70** |
| TWD (ClusterTree) | Softmax | 69.15 | 62.33 | 24.47 | 74.87 |
| | SEM | 72.88 | 63.82 | 22.55 | 77.47 |
| | AF | 70.40 | 63.28 | 24.28 | 75.24 |
| | AF (PE) | 72.37 | 65.08 | 23.33 | 76.67 |
| | AF (DCT) | 71.95 | 65.89 | 21.87 | 77.92 |
| | Softmax + JD | 73.52 | 66.76 | **24.96** | 77.65 |
| | SEM + JD | **75.93** | **67.68** | 22.96 | **79.19** |
| | AF + JD | 73.66 | 66.61 | 24.55 | 77.64 |
| | AF (PE) + JD | 73.92 | 67.00 | 23.83 | 77.87 |
| | AF (DCT) + JD | 74.29 | 67.50 | 22.89 | 78.31 |

similarity without real-valued vector embedding when the number of classes was relatively small (i.e., STL10, CIFAR10, and SVHN). In contrast, the performance of the proposed method degraded for CIFAR100, and the results of ClusterTree are particularly poor. Because Wasserstein distance can generally be minimized even if it cannot overfit, it is natural for the Wasserstein distance to make mistakes when the number of classes is large.

Next, we evaluated Jeffrey divergence regularization. Surprisingly, simple regularization dramatically improved the classification performance for all simplicial models. These results also support the idea that the main problem with Wasserstein distance-based representation learning is mainly caused by its numerical problems. Among them, the proposed AF (DCT) + JD (TV) achieves the highest classification accuracy, comparable to the cosine similarity result, and more than ten % in point improvements from the naive implementation with the Softmax function. Moreover, all the simplicial model performances with the cosine similarity combination tended to result in a lower classification error than the combination with TWD and simplicial models. From our empirical study, we propose utilizing TWD (TV) + AF models or TWD (ClusterTree) + SEM for representation learning tasks for simplicial model-based representation learning.

## 6 CONCLUSION

This study investigated a simplicial representation employing self-supervised learning with TWD. We employed softmax, simplicial embedding (Lavoie et al., 2022), and ArcFace models (Deng et al., 2019). Moreover, to mitigate the intricacies of optimizing the L1 distance, we incorporated an upper bound on the squared 1-Wasserstein distance as a regularization technique. We empirically evaluated benchmark datasets and found that a simple combination of the softmax function and TWD performed poorly. Overall, the combination of ArcFace with DCT outperformed its consine similarity counterparts. Moreover, we found that the TWD (ClusterTree) and SEM combination yielded favorable performance.

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

# A  APPENDIX

## A.1  PROOF OF PROPOSITION 1

**(Proof)** Let $B \in \{0,1\}^{N \times N_{\text{leaf}}} = [b_1, b_2, \ldots, b_{N_{\text{leaf}}}]$ and $b_i \in \{0,1\}^N$. The shortest path distance between leaves $i$ and $j$ can be represented as (Yamada et al., 2022)

$$d_{\mathcal{T}}(e_i, e_j) = w^\top (b_i + b_j - 2b_i \circ b_j).$$

$d_{\mathcal{T}}(e_i, e_j)$ is represented by a linear function with respect to $w$ with given $B$ and the constraint of $w$ and $\Pi$ are convex. Thus, we have the following representation by the Minimax theorem (v. Neumann, 1928; Fan, 1953):

$$\text{RTWD}(\mu, \mu') = \frac{1}{2} \max_{w \text{ s.t. } B^\top w = 1 \text{ and } w \geq 0} \min_{\Pi \in U(a, a')} \sum_{i=1}^{N_{\text{leafs}}} \sum_{j=1}^{N_{\text{leafs}}} \pi_{ij} w^\top (b_i + b_j - 2b_i \circ b_j)$$

$$= \frac{1}{2} \max_{w \text{ s.t. } B^\top w = 1 \text{ and } w \geq 0} \| \text{diag}(w) B(a - a') \|_1,$$

where we used $\text{TWD}(\mu, \mu') = \min_{\Pi \in U(a,a')} \sum_{i=1}^{N_{\text{leafs}}} \sum_{j=1}^{N_{\text{leafs}}} \pi_{ij} d_{\mathcal{T}}(x_i, x_j) = \| \text{diag}(w) B(a - a') \|_1$.

Without loss of generality, we consider $w_0 = 0$. We first re-write the norm $\| \text{diag}(w) B(a - a') \|_1$ as

$$\| \text{diag}(w) B(a - a') \|_1 = \sum_{j=1}^{N} w_j \left| \sum_{k \in [N_{\text{leafs}}], k \in de(j)} (a_k - a'_k) \right|,$$

where $de(j)$ is the set of descendants of node $j \in [N]$ (including itself). Using the triangle inequality, we have

$$\| \text{diag}(w) B(a - a') \|_1 \leq \sum_{j=1}^{N} w_j \sum_{k \in [N_{\text{leafs}}], k \in de(j)} |a_k - a'_k|$$

$$= \sum_{k \in [N_{\text{leafs}}]} |a_k - a'_k| \sum_{j \in [N], j \in pa(k)} w_j,$$

where $pa(k)$ is the set of ancestors of leaf $k$. Re-writing the constraint $B^\top w = 1$ as $\sum_{j \in [N], j \in pa(k)} w_j = 1$ for any $k \in [N_{\text{leafs}}]$, we obtain that

$$\| \text{diag}(w) B(a - a') \|_1 \leq \sum_{k \in [N_{\text{leafs}}]} |a_k - a'_k| = \|a - a'\|_1.$$

The latter inequality holds for any weight vector $w$, therefore, considering the vector such that $w_j = 1$ if $j \in [N_{\text{leafs}}]$ and 0 otherwise, which satisfies the constraint $B^\top w = 1$, we obtain

$$\| \text{diag}(w) B(a - a') \|_1 = \sum_{k=1}^{N_{\text{leafs}}} |a_k - a'_k| = \|a - a'\|_1.$$

This terminates the proof of this proposition.

## A.2  PROOF OF PROPOSITION 2

**(Proof)** The following holds if $B^\top w = 1$ with a probability vector $a$ (such that $a^\top 1 = 1$).

$$1^\top \text{diag}(w) B a = 1.$$

Then, by using the Pinsker's inequality we derive the following inequalities:

$$W_{\mathcal{T}}(\mu_i, \mu_j) = \| \text{diag}(w) B a_i - \text{diag}(w) B a_j \|_1 \leq \sqrt{2 \text{KL}(\text{diag}(w) B a_i \| \text{diag}(w) B a_j)},$$

and

$$W_{\mathcal{T}}(\mu_i, \mu_j) = \| \text{diag}(w) B a_j - \text{diag}(w) B a_i \|_1 \leq \sqrt{2 \text{KL}(\text{diag}(w) B a_j \| \text{diag}(w) B a_i)},$$

Thus,

$$W_{\mathcal{T}}^2(\mu_i, \mu_j) \leq \text{KL}(\text{diag}(w) B a_i \| \text{diag}(w) B a_j) + \text{KL}(\text{diag}(w) B a_j \| \text{diag}(w) B a_i)$$

