# OpenReview forum: "An Empirical Study of Simplicial Representation Learning with Wasserstein Distance"
_ICLR.cc/2024/Conference — Submitted to ICLR 2024_

### Official Review · Reviewer_hH8o · 2023-10-27

**Soundness:** 2 fair
**Presentation:** 2 fair
**Contribution:** 2 fair
**Rating:** 3
**Confidence:** 3

**Summary:**

The authors investigats obtaining simplical representation by self-supervised learning using tree-Wasserstein distance.
The authors conducted experiments on four datasets (STL10, CIFAR10, CIFAR100, and SVHN) with different combinations of the type of output layer of neural network for embedding (e.g., softmax and ArcFace) and the type of TWD.
The authors further proposed a regularization term to avoid the learning intricacies that derive from using the l1 distance in TWD.

**Strengths:**

Self supervised learning using TWD instead of cosine similarity sounds interesting.
It is plausible that metrics could affect SSL performance, and it is worth investigating the relationship between distances including TWD and the performance.

**Weaknesses:**

1. The authors say that the performance of the proposed method degrades on datasets with a large number of classes, such as CIFAR100. If this is true, the applicability of the proposed method is strongly limited.
2. The experiments are not sufficient as an empirical study. Hyperparameters such as batch size, learning rate, temperature parameter, and regularization strength are fixed, and the effects of changing these parameters are not investigated or discussed. In addition, the authors present only the average of three trials with no statistical information.

**Questions:**

1. Could you please explain why the proposed method does not work for CIFAR100? Could you provide a more detailed discussion of why minimizing the Wasserstein distance naturally fails if the number of classes is large? Why does performance degrades as the number of classes increases, even though label information is not used in self-supervised learning?
2. Since there is randomness in the learning, please provide statistical information. For example, please provide the standard deviations.
3. Could you discuss the difference between the proposed regularization and the learning rate? As the authors point out, the proposed regularization term is an upper bound on TWD (i.e., the first term in the loss). Instead of the regularization, wouldn't increasing the learning rate (or adding $\lambda \mathcal W_{\mathcal T}(a_i^{(1)}, a_j^{(2)})$) yield similar results?
4. There are $U(a, a')$ and $U(\mu, \mu')$ in the paper. What is the difference between them?
5. What are "the first term of the infoNCE" and "the second term" in Section 3.1?
6. What is $N$ in the proposed loss function in Section 4.2? Is $N$ the same as $R$ (i.e., batch size)?
7. In Section 3.3, what is $\mathbf I$? It seems that $\mathbf I$ is not a square matrix since $\mathbf B \in \\{0,1\\}^{N_\text{nodes}\times \text{leaf}}$.
8. In Section 3.3, what is $N$ in "all the edge weights $w_1 = w_2 = \ldots = w_N = \frac{1}{2}$"? (There is $w_0$ in appendix A.1.)

---

### Official Review · Reviewer_hLXp · 2023-10-31

**Soundness:** 2 fair
**Presentation:** 3 good
**Contribution:** 1 poor
**Rating:** 3
**Confidence:** 3

**Summary:**

- The authors propose to employ TWD for measuring similarities in SSL frameworks.
- Outputs from the backbone (e.g., Resnet18) is mapped on a probability simplex, then TWD is used to measure the similarity. Thanks to a good property of TWD, the training gets efficient.
- Also, they found that regularization with Jeffrey divergence is crucial for performance improvement.
- They have conducted classification experiments on several image datasets (STL10, CIFAR10, CIFAR100 and SVHN)

**Strengths:**

- The authors investigated a new loss function based on TWD in simplicial SSL. The loss function is easy to compute thanks to a good property of TWD, which allows us to incorporate into the existing simplicial SSL frameworks.
- Proposed RTWD and showed the equivalence between RTWD and the total variation. As a result, RTWD does not depend on the tree structure, which is an interesting result apart from the application to SSL.
- The paper writing is clear and easy to follow.

**Weaknesses:**

- Experimental justification is weak to support the importantce and effectiveness of TWD in SSL problem settings.
  - The authors investigated the proposed method using only Resnet18 backbone. This raises a concern of robustness of the method. Authors may want to conduct experiments other backbones at least like Resnet50 (as in Lavoie et al. 2022).
  - Similarly, I have concerns whether the proposed method works well when it is applied to other SSL methods like BYOL besides SimCLR. This is also important to show the wide applicability of the proposed method.
  - Finally, in Table 1, even if the cosine similarity is used, JD regularization is also applicable for simplicial models. It seems unfair since JD is not used in the cases of cosine similarity + simplicial models (softmax, SEM, AF(DCT)). I cannot determine where the improvement comes from (i.e., from TWD or JD). Without JD, TWD seems to be worse than (or comparable to) the cosine similarity in Table 1.
- I could not understand why Wasserstein distance (and its variants) is important in SSL setting. It seems a simple replacement of loss function without any justification. See my question below.

**Questions:**

- The authors may want to provide qualitative (or theoretical) explanations why (tree-)Wasserstein distance is better than other similarity functions (e.g., cosine similarity or other similarities between probability distributions) in the SSL setting. Of course, the Wasserstein distances have good properties as mentioned in the related work section, but I could not understand why those advantages benefit SSL. To be more specific, the authors argue that the benefits of Wasserstein distance are (1) the ability of handling different supports, and (2) the ability to identify matches between data samples, but these advantages are seemingly not utilized effectively in the proposed method, because the similarity is defined for two categorical distributions with the same support.

---

### Official Review · Reviewer_dnmG · 2023-10-31

**Soundness:** 2 fair
**Presentation:** 2 fair
**Contribution:** 2 fair
**Rating:** 3
**Confidence:** 3

**Summary:**

As enonced, the paper investigates the combination of a probability model such as Simplical Embedding, ArcFace and extensions of ArcFace with a similarity measure based on the Tree-Wasserstein distance or an upper bound distance given by a Jeffrey divergence in a self-supervised learning context. The main experiment compares 3 similarity functions (Cosine Similarity, -TWD -TV, -TWD -Cluster Tree) for classification with different simplicial models on STL10, CIFAR10, CIFAR100, SVHN with a 50-NN classifier.

The WTDs are used here to modify the similarity function component of the SimCLR SSL model, which is originally a cosine similarity. The different simplical embeddings are assumed to be learned after the encoder instead of a softmax layer.

Among the contributions: a robust version of TWD is presented and a Jeffrey divergence regularisation.

**Strengths:**

-  The paper combines very different topics (Simplical Embeddings, Tree Wasserstein Distance)

**Weaknesses:**

- The experiments are carried out with a k-NN of fixed value (50) for all databases: the choice of hyperparameter is always of great importance and can vary greatly depending on the databases and methods. The choice of 50 for all databases seems to be different from other works (see Caron et al. Emerging Properties in Self-Supervised Vision Transformers) and should be justified. Also, doing both linear and k-NN probing would have led to a better comparison with other work.

There is some confusion with regard to at least two of the referenced papers
- I assume that the context of the work is Self-Supervised Learning, where similarities between examples are used to train the representation. In this context, Simplical Embeddings can be combined with any other method and consists more of a constrained representation that could be used for a given layer of a network.  (Lavoie et al. ICLR'23 Simplicial Embeddings in Self-Supervised Learning and Downstream Classification. Note that the bibliography should cite the ICLR version). In the paper, the experiments, while not entirely similar, achieve around 70% on CIFAR 100, combining SEM with various SSL methods, and are done with linear probes.

- In the original ArcFace paper, Arface is explicitly a loss function. Even when translated into a layered form, the ArcFace model originally includes an additive angular penalty that is not present in the representation of the method.

**Questions:**

- What is the advantage of using TWD over other Wasserstein distances beyond the close form?

-What is the disadvantage of using TWD over other Wasserstein distances beyond the  weights ?

- The quality of the representations learned by SSL is often evaluated in terms of discriminability and transferability. How does changing the similarity functions affect transferability?

---

### Official Review · Reviewer_pYGX · 2023-11-01

**Soundness:** 3 good
**Presentation:** 3 good
**Contribution:** 2 fair
**Rating:** 3
**Confidence:** 3

**Summary:**

The paper empirically investigates the simplicial representation learning  with tree-Wasserstein distance (TWD) in the context of self supervised learning (SSL). In the paper, the considered TWD amounts essentially in the Total Variation (TV) metric between two simplicial embeddings issued by a neural network. Different probability models, namely softmax function, simplicial embedding, ArcFaceModel endowed with positional encoding and discrete cosine transform are discussed to achieve the simplicial representation. As the TV metric is not differentiable, the paper adjuncts a Jeffrey-Divergence regularization to the objective function of SSL to stabilize the optimization. Empirical evaluations on image classification tasks are provided to support the utility of the TWD in SSL.

**Strengths:**

- Overall, the paper is fairly well written. It  reviews the background of self-supervised learning (SSL), the optimal transport and Wasserstein distance along with the different variants of Wasserstein distance including Sliced Wasserstein distance, tree-Wasserstein distance (TWD). The link between TWD and TV metric is stated with the related restrictive assumptions. The different schemes to model the sought simplicial representations are also well exposed with the related simplifications (especially to set $\boldsymbol{K}$ instead of learning it in the ArcFace probability model). The integration of the TWD metric in the SSL is exposed. The rationale  behind adding a regularization term to the SSL objective function pertains to the non-differentiable nature and is clearly explained.

- Usefulness of the simplicial representations is studied through empirical evaluations. Image classification task is considered on four datasets. Using a KNN classifier applied to the learned simplicial representations, the paper demonstrates that some combinations of the tee-Wasserstein metric and probability models allow to obtain enhanced classification performances.

-  Overall the contribution of the paper is fairly interesting. Its originality resides in combining tree-Wassesrtein distance with simplicial representation for self supervised learning.  The empirical findings are interesting but not strong enough.

**Weaknesses:**

- The rationale behind learning simplicial representations in lieu of the customary vector representation is not well motivated in the paper.
- The paper should report not only the average classification scores but also the standard deviations. This may help appreciate the significance of the obtained results.
- As the main contribution of the paper is in the empirical study of simplicial representations using TWD, it would be a good think to provide an ablation study. For instance the hyper-parameters $\tau$ and $\lambda$ involved in the regularized objective function or the parameters $L$, $V$ used in the simplicial embedding model are fixed in the paper. Their impact on the final reported performances is not clear. Also, the evaluation on other types of applications would provide more insights on the versatility of the proposed approach.
-  Good generalization ability of the combination of probability models and TWD seems to be limited when the number of classes (CIFAR 100) grows. The paper should discuss more such behavior of the method.
- Authors should check the paper to fix some mathematical notations and typos.

**Questions:**

- How one can explain that the simplicial representation combined with the TWD performs better on the considered datasets?
- Can the authors elaborate more on the TreeCluster? The paper lacks precision on the way such this tree is built in the considered applications.
- What are the state-of-art performances of self supervised learning on the four datasets?
- Why choosing a KNN classifier? Can the authors justify that choice?

---

### Author Response · Authors · 2023-11-19
**Thank you so much for your constructive comments!**

Dear reviewers,

Thank you so much for handling the paper! We appreciate your constructive comments.
We found that some comments are from misunderstanding. However, most of them are reasonable and would like to update the paper based on that in the near future.

Thanks!

Authors

---

### Meta-Review · Area_Chair_GugN · 2023-12-12

**Metareview:**

Overall, the paper received a fairly negative and uniform evaluation from all reviewers, who pointed out various issues with presentation and experimental setup. We hope these reviews are useful for the authors to improve their draft.

**Justification For Why Not Higher Score:**

clear reject

**Justification For Why Not Lower Score:**

low score reflects accurately limitations, no need to go further below

---

### Decision · Program_Chairs · 2024-01-16

Reject